# `BlockScan`: Detecting Anomalies in Blockchain Transactions

**Jiahao Yu**[5]* **Xian Wu**[2]* **Hao Liu**[3,4]† **Wenbo Guo**[1] **Xinyu Xing**[4,5]†

[1] UC Santa Barbara  [2] Meta AI  [3] New York University  [4] sec3  [5] Northwestern University

## Abstract

We propose `BlockScan`, a customized Transformer for anomaly detection in blockchain transactions. Unlike existing methods that rely on rule-based systems or directly apply off-the-shelf large language models (LLMs), `BlockScan` introduces a series of customized designs to effectively model the unique data structure of blockchain transactions. First, a blockchain transaction is multi-modal, containing blockchain-specific tokens, texts, and numbers. We design a novel modularized tokenizer to handle these multi-modal inputs, balancing the information across different modalities. Second, we design a customized masked language modeling mechanism for pretraining the Transformer architecture, incorporating RoPE embedding and FlashAttention for handling longer sequences. Finally, we design a novel anomaly detection method based on the model outputs. We further provide theoretical analysis for the detection method of our system. Extensive evaluations on Ethereum and Solana transactions demonstrate `BlockScan`'s exceptional capability in anomaly detection while maintaining a low false positive rate. Remarkably, `BlockScan` is the only method that successfully detects anomalous transactions on Solana with high accuracy, whereas all other approaches achieved very low or zero detection recall scores. This work sets a new benchmark for applying Transformer-based approaches in blockchain data analysis.

## 1 Introduction

Together with the emergence of blockchain techniques comes different attacks against Decentralized Finance (DeFi) protocols, such as double-spending [1], partition [2], and front-running [3]. These attacks seriously threaten the asset security of billions of blockchain users. Mitigating these attacks requires detecting anomalous transactions that indicate potential malicious behavior or misuse of DeFi protocols. In this work, we define *anomalous transactions* as transactions that significantly deviate from normal, benign usage patterns and often arise from exploiting smart contract vulnerabilities. These anomalies are characterized by irregular or suspicious method calls, abnormal numerical parameters, and unforeseen sequences of operations inconsistent with typical user behavior. By identifying such outliers at runtime, DeFi platforms can issue early warnings and trigger interventions to prevent or at least minimize asset losses [4].

Prior works in anomalous transaction detection predominantly rely on either rule-based or traditional ML methods. Rule-based approaches cannot generalize to unseen patterns. Traditional ML models [5, 6] like Gaussian Mixture Models (GMM) and Long Short-Term Memory (LSTM) cannot capture the complex, high-dimensional temporal dependencies inherent in blockchain transaction data due to limited capabilities. More recently, BlockGPT [7] harnesses a GPT model for anomaly detection. However, vanilla GPT models cannot capture their unique data attributes and do not have an effective anomaly detection mechanism.

---

*Equal contribution, the order of authors is decided by flipping of a coin.

†Work done while at sec3.

39th Conference on Neural Information Processing Systems (NeurIPS 2025).

To overcome these limitations, we propose `BlockScan`, a customized transformer model specifically tailored for detecting anomalous transactions in DeFi. `BlockScan` uses a BERT-style Transformer architecture with mask language modeling (MLM). Rather than focusing on text generation, we aim to learn robust representations of typical on-chain behaviors and identify deviations from normal usage. By masking and reconstructing transaction tokens, `BlockScan` assigns higher reconstruction errors to suspicious or atypical patterns, thereby facilitating accurate anomaly detection.

A significant challenge in developing a transformer-based model for blockchain data is the **multi-modal and complex nature of transactions**. Each transaction typically involves: 1) *Blockchain-Specific Tokens (Addresses/Hashes)* that are often unique and long hexadecimal strings, making a naive one-hot encoding impractical due to the explosion in vocabulary size. 2) *Large Numerical Values*: Transaction amounts (*e.g.,* value and gas price) can be very large, easily spanning dozens of digits. Also, there are mixed types of numbers in a transaction, such as decimal numbers and hexadecimal numbers. 3) *Textual Fields*, which are used in transaction logs. Balancing vocabulary size and preserving critical information is thus non-trivial. To address this challenge, `BlockScan` employs a novel tokenization scheme that integrates distinct strategies per modality. For addresses and hashes, we preserve only frequently occurring ones as single tokens to balance vocabulary size and information retention. For numerical values, we first convert decimals to hexadecimal format for consistency and information density, then apply subword tokenization with a fixed vocabulary size. For textual data like log messages, we utilize the subword tokenizer. This multi-modal tokenization approach enables `BlockScan` to effectively process transaction data while maintaining a manageable vocabulary and preserving essential information. Furthermore, we leverage RoPE and FlashAttention to modify the base BERT model such that `BlockScan` can handle long inputs. We train our transformer model on a large dataset of benign transactions to learn the patterns of normal transactions, then feed the trained model with masked transactions and calculate how well the model can reconstruct the transaction. We use the reconstruction errors as the metric for anomaly detection. We further provide a *theoretical analysis* on the error bound under the natural assumption, showing that anomalous transactions will have a higher error bound.

We evaluate `BlockScan` on real-world transactions from Ethereum and Solana networks, comparing it against four baselines: rule-based, traditional ML-based, BlockGPT, and GPT-4. `BlockScan` demonstrates superior accuracy and lower false positive rates compared to existing methods. We further validate our key designs and hyper-parameter sensitivity through an ablation study. The code, model, and datasets are available at an anonymous link. [3] `BlockScan` is the first open-source and best-performing transformer-based anomaly detection for DeFi that provides a theoretical guarantee.

## 2 Background

**Blockchain.** Blockchain is a decentralized, distributed ledger technology that enables secure and tamper-resistant record-keeping. Originally developed for Bitcoin [8], it achieves consensus across a network through mechanisms like Proof of Work (PoW) [9] or Proof of Stake (PoS) [10]. Its immutability and transparency make it suitable for various applications beyond cryptocurrencies, including supply chain management, healthcare, and finance.

**Smart Contracts and Transactions.** Smart contracts are self-executing agreements encoded on blockchain platforms like Ethereum, enabling automated transactions within decentralized applications (DApps). These contracts process conditions, manage assets, and update the ledger autonomously. While smart contracts enhance transparency and security in DeFi and other applications, their immutability after deployment makes vulnerability detection crucial. This work focuses on identifying anomalous transactions that deviate from typical patterns, which could indicate vulnerabilities in contract logic or implementation. Early detection of such anomalies is essential for preventing potential losses [11, 12].

## 3 Related Work

**Intra-Transaction Anomaly Detection** This category encompasses methods that modeling the internal structure (*e.g.,* function calls, parameters, logs, addresses) within a single blockchain transaction to identify deviations from normal behavior. `BlockScan` belongs to this category. A closely related

---

[3] https://github.com/nuwuxian/tx_fm

approach is BlockGPT [7], which also utilizes Transformer architectures, specifically a GPT-like causal language model, to predict subsequent tokens in a flattened transaction trace. However, this approach faces several key limitations: transactions do not naturally form sequential data like language, making next-token prediction less meaningful, and the tokenization method (rounding numerical values to avoid vocabulary explosion) can obscure critical transaction details. `BlockScan` introduces a custom, multi-modal tokenizer to address this, whereas approaches like BlockGPT might simplify or lose information. Some approaches [13] alternatively use existing LLMs like ChatGPT without fine-tuning, feeding them raw transaction data, but these methods are limited by both input length constraints and the model's knowledge boundaries.

Besides transformer-based methods, non-transformer-based methods for anomalous transaction detection comprise two main categories: traditional ML and rule-based approaches. The first category employs conventional models like Gaussian mixture models to estimate transaction density [5], flagging low-density transactions as anomalous. However, these methods heavily depend on the quality of transaction features used for representations, limiting their generalizability. Heuristic-based approaches, such as those detecting anomalies through sequence length analysis [7], make simplified assumptions about transaction patterns. While straightforward to implement, these heuristics are often too rigid and can be easily circumvented by adversaries who don't conform to expected patterns.

**Inter-Transaction and Local Network Anomaly Detection** Moving beyond individual transactions, a higher level anomaly detection analyzes a set of transactions or entities (*e.g.,* smart contracts) to detect suspicious interactions. For instance, BERT4ETH [14] applies BERT-like pre-training to transaction histories for account-level fraud detection, which tackles the different scope from our focus. Graph Neural Networks (GNNs) are another prominent technique in this space [15, 16, 17, 18]. They model blockchain data as a graph, where transactions and addresses can be nodes, and the flow of value or calls between them form edges. GNNs are effective at learning representations of nodes based on their local neighborhood and can identify anomalies like small money laundering rings, unusual interaction sequences between a few contracts [15]. While powerful for capturing these localized relational anomalies, GNNs often abstract the fine-grained internal details of each transaction that low-level approaches like `BlockScan` focus on. The insights from inter-transaction analysis could potentially serve as a complementary signal to `BlockScan`'s intra-transactional findings.

**Chain-level Temporal Anomaly Detection** The higher level of anomaly detection shifts the focus on analyzing network-wide statistics and temporal sequences of blockchain activity. This involves monitoring time-series data derived from the blockchain, such as the total transaction volume per unit of time, average gas prices, the creation rate of new smart contracts or the frequency of specific global events [19, 20]. Anomalies at this level might manifest as sudden, unexpected spikes or drops in these global metrics, indicating large-scale events like network congestion, widespread exploitation of a common vulnerability, or coordinated market manipulation. Various time-series analysis techniques can be employed, ranging from classical statistical models (*e.g.,* Autoregressive Integrated Moving Average [21]) to machine learning models like LSTM [22]. While these macro-level and temporal approaches are crucial for understanding the overall health and security trends of a blockchain ecosystem, they generally do not provide insights into the maliciousness of a specific, individual transaction's internal logic, which is the primary objective of `BlockScan`.

## 4 Key Techniques

### 4.1 Technique Overview

**Tokenizer.** As demonstrated in Figure 1, a blockchain transaction mainly consists of three types of inputs: function and address signature in hash values, function logs in natural languages, and function arguments in numbers. This hybrid data type makes a transaction naturally to be multi-modal. As such, directly applying existing tokenizers designed for language models to blockchain transactions will be problematic. First, existing tokenizers will treat hash values as numbers and divide them into sub-tokens. However, these numbers themselves are meaningless, instead, they are just used to represent different entities. Second, the numbers in blockchain transactions have a very large value range and large values frequently show up. Directly applying the existing tokenizers will divide a large number into many sub-tokens and thus result in ultra-long sequences for individual transactions. To solve the first issue, we use one-hot tokenization for hash values. We only consider the top 7,000 frequent hash values in our training dataset and treat the rest as "OOV" (Out of Vocabulary). This

```
1. "type": "CALL",
2. "from": "0xc1f351...5d0",
5. "gas": 1962908,
3. "to": "0x4deca5...bac",
4. "func": "0x9fa0bc94",
7. "args": [...],
   "output": [{"type": "data",
8.     "data": "0x000000...009"}],
   "calls": [{
        "type": "DELEGATECALL",
        "from": "0x4deca5...bac",
        "gas": 1930278,
        "to": "0x35dd16...5e8",
10.     "func": "0x9fa0bc94",
        "args": [...],
        "output": [...],
        "calls": [...],
        "logs": [...]
   }]
   "logMessages": [
9. "Program PhoeNi... invoke [2]",
   "Program PhoeNi... consumed
          none compute units"],
6. "value": 0
```

[START]   [CALL]   0xc1f351...5d0   0x4deca5...bac   0x9fa0bc94
start indicator   call indicator   from address   to address   function id
of the calling   1.   2.   3.   4.

0x000000...39c   0x000000...000   [INs]   data   0x476f76...000
5. gas 1962908   6. value 0   7. input   7.
converted to hex   converted to hex   indicator   input type and data

address   0x000000...5d0   data   ......   [OUTs]   data
7. input type and data   8. output indicator

0x000000...009   [logs]   "Program  PhoeNi...units"   [END]
8.   9. log   9.   end indicator
output type and data   indicator   log messages   of the calling

[START]   [DELEGATECALL]   [OOV]   0x35dd16...5e8   0x9fa0bc94
10.   out of vocabulary
subsequent call's infomation

0x000000...426   [NONE]   [INs]   data   0x476f76...000   ......   [OUTs]
data does not exist

data   0x000000...009   [END]   [START]   [STATICCALL]   ......   [END]

Figure 1: **Tokenizer of** `BlockScan`. `BlockScan` tokenizes transactions by flattening nested JSON using depth-first search, assigns unique tokens to frequent addresses while marking infrequent ones as "OOV", and uses special tokens ("[START]", "[END]", "[Ins]") to mark function artifacts.

method can constrain the vocabulary size, which helps reduce model parameters and improve training efficiency. We further train our own number tokenization model to handle numbers. Different from existing tokenizers, our model can better tailor to the large numbers in blockchain transactions and give shorter token sequences for large numbers. Finally, we still apply the text tokenizer to function logs to capture their semantic meanings. As demonstrated in §6, our customized tokenizer is critical for learning foundation models and final anomaly detection.

**Model design.** We make a different design choice from BlockGPT [7] and use a BERT structure together with MLM for our foundation model. The key rationale is to reduce training complexity and improve overall training efficiency. Specifically, we do not need to generate new transactions, and training GPT models are in general more difficult than BERT as predicting the future without any context is harder than filling missing parts with certain context. Besides, our main focus is to learn patterns of normal transactions. Thus, we select MLM, which provides enough pattern-learning capabilities. We choose to apply RoPE embedding and FlashAttention in our model to handle long input sequences. The reason we choose this technique combination rather than other ones like LongLoRA [23] is to consider computational cost and algorithmic simplicity. These techniques still keep a one-stage pretraining is simpler and more efficient than two-stage training, which is required by LoRA-based approaches.

**Post-training detection.** With a trained foundation model, we feed masked transactions into the model and use reconstruction error as the metric for identifying abnormal transactions. This approach is effective because the model learns patterns of normal transactions, causing anomalous ones to exhibit higher reconstruction errors due to their irregularity. While we explored alternative approaches using transaction embeddings and one-class contrastive learning [24] with both <CLS> token and full token embeddings (detailed in §D.1), none outperformed the simple reconstruction error approach. Therefore, we adopt this straightforward method, which provides both superior detection performance and minimal computational overhead.

## 4.2   Tokenization

To address these challenges in tokenization we discussed above, `BlockScan` introduces a custom tokenizer specifically designed for the unique multi-model characteristics of blockchain transaction data. We first flatten the raw JSON data into a sequence of function calls and apply a depth-first search to track function callings. We use "[START]" and "[END]" tokens to help the model identify the beginning and end of each function call within the sequence. Additionally, "[Ins]" and "[OUTs]" tokens are used to mark the input and output arguments of functions, which can vary in number. To further distinguish between data types, we use tokens like "data" and "address" to indicate whether the argument is a data value or an address. These special tokens enable the model to clearly recognize the type and boundaries of variable-length information, improving accuracy in transaction tracing.

After pre-processing the transaction trace, we then treat unique hash addresses as individual tokens, which can significantly reduce the overall token count. Given the large number of unique addresses, we rank them by frequency and retain the top 7,000 most frequent addresses. Addresses that fall outside the top 7,000 are treated as a single "OOV" token, as shown in Figure 1. In real-world scenarios, frequent addresses are often associated with public smart contracts or exchanges, and preserving them as single tokens improves the system's ability to understand transaction behavior.

For values, as illustrated in Figure 1, there are both decimal (*e.g.,* gas) and hexadecimal numbers (*e.g.,* output data). We first convert all decimal numbers into 40-character hexadecimal format. This approach provides a more compact representation of large numbers as the hexadecimal number is more concise than the decimal number. Small numbers typically begin with "0x000...", which can be learned as a single token. Therefore, this conversion does not lead to a significant increase in token count for small numbers. The consistent formatting of values across all transactions simplifies processing and comprehension for models.

Unlike hash addresses, log messages often convey information about the same object, such as program status, across different function calls. For example, in Figure 1, log messages like "Program PhoeNi invoke [2]" and "Program PhoeNi consumed none compute units" vary in details but relate to the same event. Treating each log message as a unique token would obscure relationships between messages, which often share common topics. Subword tokenization preserves these connections, ensuring that the model can recognize similarities across different log messages.

The token dictionary size is set at 30,000 to balance the trade-off between token count and information granularity. After allocating space for special tokens and preserved hash address tokens, we apply WordPiece tokenization to learn on the remaining tokens for numbers and log messages.

## 4.3 Model Design

`BlockScan` adapts the RoBERTa model [25] to train an auto-encoder specifically for transaction tracing. `BlockScan` employs a MLM strategy, where $g\%$ of tokens in each transaction are randomly masked. The model is then trained to reconstruct the original transaction from the masked tokens, learning robust representations in the process. However, tokenized transaction data can be significantly longer than typical natural language sequences, posing additional challenges during training. To address these challenges, we incorporate two key techniques: (1) We replace the absolute position embeddings in RoBERTa with Rotary Position Embeddings (RoPE) [26], which provide more efficient handling of long-range dependencies. (2) We leverage *FlashAttention* [27] to accelerate the attention mechanism, improving memory efficiency and reducing computational overhead, making it feasible to train on long transaction sequences.

## 4.4 Post-training Detection

After training, we can deploy `BlockScan` for detecting anomalous transaction sequences. The motivation behind applying `BlockScan` for transaction anomaly detection is that since the model is trained on benign transaction sequences, it can accurately predict masked tokens if the sequence is also benign. Hence, the anomalous score of a transaction can be derived based on the prediction results on the masked tokens. Specifically, for a given transaction, we randomly mask a ratio of the tokens, similar to the training process, and input the masked sequence into the trained model. The probability distribution over the possible tokens for each MASK position represents the likelihood of each token in that position. We construct a candidate set of the top-$s$ most likely tokens for each masked position. If the true token appears within the top-$s$ candidate set, we consider the token as benign. Conversely, if the true token is not in the top-$s$ candidate set, it is treated as anomalous. The reason why we do not directly predict based on the most likely token is that the addresses and values are more challenging than nature language texts to predict, and having a candidate set tolerant to the prediction error is more reasonable. After ranking the transactions by the anomalous score, we can select the top $k$ transactions with the highest anomalous score as anomalous. $k$ can be dynamically adjusted based on how the smart contract developers trade off between false positives and security of the transactions.

# 5   Theoretical Analysis

We present some theoretical insights to show that our approach is well motivated in theory. Specifically, we show that transactions deviating from the learned benign distribution tend to incur higher prediction errors under our pretrained transformer model.

We begin by introducing the notation and basic definitions. Each blockchain transaction $D$ is represented as a sequence of tokens, and $D'$ denotes a partially masked version of $D$. We train a transformer model $h$ to reconstruct $D$ from $D'$ by minimizing the cross-entropy loss $\mathcal{L}$ over the benign transaction distribution $P_{\text{benign}}$. The resulting optimal model $h^*$ is obtained by solving the following loss function: $\mathbb{E}_{(D,D')\sim P_{\text{benign}}}[\mathcal{L}(h(D'), D)]$. We then analyze the expected loss of $h^*$ when applied to malicious transactions drawn from $P_{\text{mal}}$, expressed as $\mathbb{E}_{(D,D')\sim P_{\text{mal}}}[\mathcal{L}(h^*(D'), D)]$. Leveraging results from domain adaptation theory [28, 29], we derive an upper bound on this risk.

**Theorem 5.1** ([28])**.** *The expected loss of $h^*$ on the malicious distribution $P_{mal}$ satisfies:*

$$\mathbb{E}_{(D,D')\sim P_{mal}}[\mathcal{L}(h^*(D'), D)] \leq \mathbb{E}_{(D,D')\sim P_{benign}}[\mathcal{L}(h^*(D'), D)] + \frac{1}{2}d_{\mathcal{H}\Delta\mathcal{H}}(P_{benign}, P_{mal}) + \lambda, \quad (1)$$

*where the first R.H.S. term is the benign loss, $d_{\mathcal{H}\Delta\mathcal{H}}$ is the $\mathcal{H}\Delta\mathcal{H}$-divergence between $P_{benign}$ and $P_{mal}$, and $\lambda$ is a constant representing the minimum joint error of any hypothesis in $\mathcal{H}$.*

In Theorem 5.1, the $d_{\mathcal{H}\Delta\mathcal{H}}(P_{\text{benign}}, P_{\text{mal}})$ term measures the dissimilarity between the distributions of partially observed benign and malicious transactions. A larger divergence increases the upper bound on the loss for malicious data. Consequently, malicious transactions are expected to yield higher prediction losses (*i.e.,* our anomaly scores) than benign ones, justifying their detection by ranking these scores.

# 6   Experiments

## 6.1   Experimental Setup

**Dataset.** We primarily focus on Ethereum and Solana transactions in our experiments. We sample transactions from interactions with 5 DeFi applications for Ethereum and 10 applications for Solana to ensure diverse transaction patterns. For each DeFi application, transactions are ordered by their block timestamps and split into 80% for training and 20% for evaluation as benign transactions. This per-application sequential split is crucial to prevent time travel data leakage, ensuring that the model is trained exclusively on past data without access to future information. Such a methodology can maintain the integrity of performance metrics by avoiding artificially inflated results that could arise if the model inadvertently learned from future transactions.

Specifically, the Ethereum dataset is collected from October 2020 to April 2023, while the Solana dataset is collected from September 2023 to December 2023. For malicious transactions, they are also sampled from these DeFi applications and occurred after the sampling periods of benign transactions. This approach guarantees that the model is trained solely on known benign transactions up to the cutoff dates, preventing any inadvertent exposure to future anomalous patterns during training. We sample these malicious transactions from verified transaction vendors, including Zengo, TRM Labs, and CertiK. Moreover, we manually verify the malicious nature of these transactions to ensure the quality of the dataset. For both benign and malicious transactions, we manually clean them to remove transactions unrelated to the target applications or failed transactions. We provide more details on the dataset in Appendix C. As far as we know, this is the open-sourced dataset for transformer-based blockchain transaction anomaly detection.

**Evaluation Metrics.** We adopt the evaluation methodology from BlockGPT [7], where transactions are ranked based on their detection scores produced by the models. Specifically, the top-$k$ transactions with the highest scores are labeled as anomalous, while the remaining transactions are classified as benign. The binary classification performance is evaluated using the following metrics: *False Positive Rate* (FPR), *Recall*, and *Precision*. In our experiments, we select $k$ values from the set 5, 10, 15 for Ethereum and 10, 15, 20 for Solana considering the number of collected to evaluate the model's performance at different detection thresholds. A larger $k$ value indicates a higher detection threshold, potentially leading to more false positives but could detect more anomalous transactions, which can

| Method | k=10 | | | k=15 | | | k=20 | | |
|---|---|---|---|---|---|---|---|---|---|
| | **FPR** | **Recall** | **Precision** | **FPR** | **Recall** | **Precision** | **FPR** | **Recall** | **Precision** |
| **BlockGPT** | 0.47% | 16.67% | 30% | 0.73% | 22.22% | 26.67% | 1% | 27.78% | 25% |
| **Doc2Vec** | 0.67% | 0% | 0% | 1% | 0% | 0% | 1.13% | 0% | 0% |
| **GPT-4o** | 0.67% | 0% | 0% | 1% | 0% | 0% | 1.13% | 0% | 0% |
| **Heuristic** | 0.67% | 0% | 0% | 1% | 0% | 0% | 1.13% | 0% | 0% |
| BlockScan | **0.13%** | **44.44%** | **80%** | **0.2%** | **66.67%** | **80%** | **0.47%** | **72.22%** | **65%** |

Table 1: **Performance comparison with different $k$ values for Solana.**

| Method | k=5 | | | k=10 | | | k=15 | | |
|---|---|---|---|---|---|---|---|---|---|
| | **FPR** | **Recall** | **Precision** | **FPR** | **Recall** | **Precision** | **FPR** | **Recall** | **Precision** |
| **BlockGPT** | 0.14% | 40% | 80% | 0.42% | 70% | 70% | 0.99% | 80% | 53.33% |
| **Doc2Vec** | 0.56% | 10% | 20% | 1.12% | 20% | 20% | 1.83% | 20% | 13.33% |
| **GPT-4o** | 0.28% | 30% | 60% | 0.98% | 30% | 30% | 1.55% | 40% | 26.67% |
| **Heuristic** | 0.14% | 40% | 80% | 0.42% | 70% | 70% | 1.13% | 70% | 46.67% |
| BlockScan | **0%** | **50%** | **100%** | **0.28%** | **80%** | **80%** | **0.97%** | **80%** | **53.33%** |

Table 2: **Performance comparison with different $k$ values for Ethereum.**

be varied based on how the DeFi owner wants to trade off between false positives and security. We run three independent runs for each experiment and report the average results.

**Model architecture and hyper-parameters.** We use the BERT-base architecture, which includes 100 million parameters, for training the Ethereum task, and the BERT-large architecture, with 300 million parameters, for training the Solana task. We set the learning rate to 5e-5 and use a batch size of 32 for the Ethereum task and 4 for the Solana task, respectively. For the Ethereum task, the maximum sequence length is set to 1,024 tokens, while for the Solana task, we increase the maximum sequence length to 8,192 tokens to accommodate the longer transactions. Please refer to §D.1 for a detailed setup of the training hyper-parameters for both datasets. In the inference phase, we set the mask ratio $g$ to 15% and the number of candidate tokens $s$ is set to 3 on both datasets.

**Baselines.** To evaluate the effectiveness of BlockScan, we compare it against several anomalous transaction detection methods: ❶ **BlockGPT** [7]: It pretrains a causal transformer model on the transaction corpus to learn typical benign transaction patterns. The underlying intuition is that anomalous transactions deviate from these learned patterns and are therefore difficult to predict. BlockGPT calculates the sum of the conditional log-likelihoods for each token in a transaction sequence, with lower likelihoods indicating potential anomaly. The top-$k$ transactions with the lowest scores are flagged as anomalous. ❷ **Doc2Vec** [30]: It represents the transaction as a bag of words and leverages the distributed representation of words to represent the transaction. These vectorized transactions are then analyzed using a GMM to estimate the probability of each transaction being anomalous. This probabilistic approach allows for the identification of anomalous transactions based on their likelihood within the learned distribution. ❸ **GPT-4o**: This method utilizes a state-of-the-art commercial language model to assign an anomaly score ranging from 0 to 100 to each transaction. This approach relies on the extensive pre-training of the language model, which could potentially encompass a wide variety of anomalous transaction patterns, enabling it to detect suspicious activities based on learned knowledge. ❹ **Heuristic-based methods**: Previous study [31] has highlighted that machine learning models can sometimes achieve decent detection rates by leveraging trivial features like input length in detection tasks. To explore this, our heuristic-based approach uses the length of a transaction as the sole feature, operating under the assumption that anomalous transactions are typically longer than benign ones.

By comparing BlockScan with these diverse baselines, we aim to demonstrate its superior performance in accurately identifying anomalous transactions while mitigating the impact of potential confounding factors present in other detection methods.

## 6.2 Experimental Results

**Comparison with Baselines.** We show the FPR, Recall, and Precision of BlockScan and other baselines in Table 1 and Table 2. As the results show, BlockScan outperforms all baseline methods

| Models | k=10 | | | k=15 | | | k=20 | | |
|---|---|---|---|---|---|---|---|---|---|
| | FPR | Recall | Precision | FPR | Recall | Precision | FPR | Recall | Precision |
| BlockScan | **0.13%** | **44.44%** | **80%** | **0.2%** | **66.67%** | **80%** | 0.47% | 72.22% | 65% |
| - Tokenizer | 0.67% | 0% | 0% | 1% | 0% | 0% | 1.33% | 0% | 0% |
| - Log message | 0.67% | 0% | 0% | 1% | 0% | 0% | 1.33% | 0% | 0% |
| - RoPE | 0.4% | 22.22% | 40% | 0.53% | 38.89% | 46.67% | 0.80% | 44.44% | 40% |
| BlockScan-100m | 0.6% | 5.56% | 10% | 0.93% | 5.56% | 6.67% | 1.27% | 5.56% | 5% |
| BlockScan-g=5 | 0.27% | 33.33% | 60% | 0.4% | 50% | 60% | 0.53% | 66.67% | 60% |
| BlockScan-g=10 | 0.27% | 33.33% | 60% | 0.4% | 50% | 60% | 0.53% | 66.67% | 60% |
| BlockScan-s=1 | 0.13% | 44.44% | 80% | 0.27% | 61.11% | 73.33% | **0.4%** | **77.78%** | **70%** |
| BlockScan-s=5 | 0.13% | 44.44% | 80% | 0.27% | 61.11% | 73.33% | 0.47% | 72.22% | 65% |

Table 3: **Ablation study on** BlockScan **for Solana.**

across various $k$ values for both the Ethereum and Solana datasets. Notably, on the Solana dataset, most baseline methods (Doc2Vec, GPT-4o, and Heuristic) consistently fail to detect any anomalous transactions, achieving a recall and precision of 0% for all $k$ values. This indicates that all transactions flagged as anomalous by these methods are, in fact, benign. While BlockGPT is able to detect some anomalous transactions, its recall and precision are significantly lower than those of BlockScan. For instance, at $k = 20$, BlockGPT achieves only a 27.78% recall with a FPR of 1%. In contrast, BlockScan detects the majority of anomalous transactions (*i.e.,* 13 out of 18) with a lower FPR of 0.47%.

We have the following potential reasons for the failure of these baseline methods: 1) Doc2Vec's approach of representing transactions as a bag of words likely fails due to its inability to capture the sequential dependencies and contextual nuances crucial for distinguishing between benign and anomalous transactions. 2) Despite its extensive pre-training, GPT-4o may underperform because it is not specifically fine-tuned for blockchain-specific anomalous transaction detection, making it less effective in identifying such domain-specific anomalies. 3) The heuristic method fails when the heuristics are not accurate for those anomalous transactions that have similar length as benign transactions. 4) BlockGPT, which shares the most similar idea with BlockScan, fails to detect anomalous transactions because the casual language model structure may not be suitable for detection task. For each token in the transaction, it only considers preceding information while BlockScan can analyze both previous and subsequent information for tokens to predict.

In contrast to baseline methods's failure, BlockScan demonstrates strong performance with significantly lower FPRs and much higher recall and precision scores, especially as the $k$ threshold increases. For example, at $k = 10$ on the Ethereum dataset, BlockScan achieves an FPR of 0.28%, a recall of 80%, and a precision of 80%, which means BlockScan can successfully detect 8 out of 10 anomalous transactions while only predicting 2 false positives.

These results highlight the effectiveness of BlockScan in accurately identifying anomalous transactions while minimizing false positives, thereby demonstrating its superiority over existing detection methods in both Ethereum and Solana environments. The baselines' failure to detect anomalous transactions also underscores the challenge of this task and the importance of leveraging advanced methods like BlockScan for robust blockchain transaction security.

**Effect of Core Components.** We conduct an ablation study on BlockScan by removing each core component individually to analyze its impact on detection performance with the Solana dataset. The first component we ablate is the *tokenizer*, which is specifically designed to handle transaction data in BlockScan. To evaluate its significance, we replace the custom tokenizer with the default WordPiece tokenizer from BERT, allowing us to observe how much this tailored tokenization contributes to the model's success. Next, we examine the effect of *log message*, which is the printed information when executing these transactions. As mentioned in §4.2, we use subword tokenization to encode the log messages in order to preserve their context information. Here, we substitute this approach by treating each log message as a unique token, similar to how we treat the hash addresses, and measure the resulting change in performance. Lastly, we study the effect of the *RoPE embedding*, which we employ to capture the relative position information between tokens. In this ablation, we replace it with the default absolute positional embedding.

The results are presented in the upper half of Table 3. From these experiments, we draw the following conclusions. First, substituting our customized tokenizer with the default BERT tokenizer, while keeping all other components unchanged, caused the model to fail to detect any anomalous

transactions (*i.e.,* the recall was 0 across different $k$ values). This underscores the importance of our customized tokenizer, as the default BERT tokenizer, trained on general text data, is unable to capture the complex structure of specific transaction traces. Second, we observed that the model also struggled to differentiate between benign and anomalous transactions when we altered the log message encoding strategy. This suggests that the log messages may contain key information about the transaction status in the Solona task, and an appropriate encoding method, such as a subword tokenizer, can extract this information effectively. Lastly, replacing our relative positional embeddings with absolute positional embeddings led to a significant drop in model performance, with a decrease in recall of nearly 20% to 30% across various $k$ values. This emphasizes the importance of relative positional embeddings for effectively handling long sequences (*e.g.,* a sequence length of 8192).

We also conduct an ablation study on the impact of FlashAttention on the training time and memory usage of `BlockScan` in Table 9. The results show that the integration of FlashAttention reduces the training time and memory usage while maintaining the detection performance. Furthermore, we investigate the selection of the base model in Table 10 by replacing the RoBERTa model with other state-of-the-art BERT-like models like DeBERTa [32]. The results demonstrate that `BlockScan` achieves consistent performance and is agnostic to the choice of base model.

**Hyper-parameters sensitivity analysis.** We further investigate the impact of key hyper-parameters and model architecture on the final model performance in the Solana task. Specifically, we introduce two additional hyper-parameters during detection phrase: the detection mask percentage $g$ and the number of candidate tokens $k$ used when calculating the mask prediction accuracy. By varying $g$ and $s$ within $\{5, 10, 15\}$ and $\{1, 3, 5\}$, respectively, we assess the model's robustness to these parameters. Additionally, While BERT-large is the default model on Solana dataset, we replace it with BERT-base to evaluate the influence of different model architectures on the final performance.

As shown in the lower half of Table 3, our model demonstrates a degree of robustness to variations in $g$ and $s$. Recall that the default values for $g$ and $s$ in `BlockScan` are 15% and 3, respectively. Notably, `BlockScan`-s=1 even outperforms `BlockScan` when $k = 20$, suggesting that a simpler set of hyperparameters can still achieve relatively good performance. However, when the model architecture is switched from BERT-large to BERT-base, a noticeable performance drop occurs on the Solona dataset. This is likely due to the dataset's large number of training samples (*i.e.,* almost 30,000) and longer sequence length (*i.e.,* 8,192 tokens), which smaller models like BERT-base struggle to handle effectively.

**Larger Dataset.** The dataset's size of malicious transactions is constrained due to the manual validation and cross-checking required to confirm their malicious nature. To address this, we perform an additional experiment using only manually verified malicious transactions on the Ethereum dataset, without cross-checking, resulting in a malicious dataset that is 10x larger. The findings, presented in Figure 2, further validate the effectiveness of `BlockScan` in detecting anomalous transactions within a larger dataset.

# 7 Limitations and Discussion

**Tokenizer and Model Adaptability.** Our approach builds the tokenizer using a large transaction corpus. To maintain detection performance, we recommend periodically rebuilding tokenizer to align with evolving transaction patterns. Our model's training on benign data enables it to learn typical patterns and detect anomalies, providing adaptability to new malicious strategies. However, if attack strategies closely mimic benign patterns, detection becomes challenging. Regular model retraining on updated data helps ensure optimal accuracy as blockchain ecosystem evolves.

**Fine-Tuning GPT-4o.** As shown in §6.2, directly using GPT-4o as a detector results in poor performance. We further fine-tune GPT-4o via OpenAI API on Ethereum dataset but still observe limited performance. We hypothesize that this is because the fine-tuning API is coarse-grained and does not allow next token prediction nor customization of tokenization, which is crucial for our task. Detailed results are shown in Table 7.

**Robustness to Noise.** In blockchain transactions, the ratio of benign to malicious transactions is typically highly imbalanced. Given this high imbalance, even if some potential malicious samples are inadvertently included in the training set, their impact is minimal, as the model predominantly learns the representation of the majority class (benign transactions). To assess the model's robustness to

noise, we conduct an experiment simulating an extreme case where half of the malicious transactions were intentionally included in the training set for the Ethereum dataset. While this scenario caused a slight drop in detection performance, the model remained effective. These results demonstrate that our approach maintains robustness to a reasonable level of noise in the data (Table 8).

# 8   Conclusion

We presented `BlockScan`, a transformer-based model designed for detecting anomalous transactions in DeFi ecosystems such as Ethereum and Solana. By leveraging MLM and carefully designed tokenization techniques, `BlockScan` efficiently handles the complexity and diversity of transaction data. Additionally, we open-sourced the code, model, and datasets, making `BlockScan` the first open-source solution for LLM-based anomalous transaction detection in DeFi. We hope that this contribution will serve as a valuable resource for the research community, facilitating further advancements in the development of scalable and robust anomalous transaction detection systems.

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

# A Broader Impact

This work focuses on anomaly detection in blockchain transactions, with the goal of mitigating risks in DeFi applications. By identifying transactions that exhibit abnormal behaviors, such as those arising from smart contract vulnerabilities, our work contributes to enhancing the security and reliability of blockchain-based systems, which play an increasingly important role in financial and other industries.

We emphasize that all transactions used in this study are publicly available and sourced from blockchain networks. No private or permissioned transaction data has been used, ensuring compliance with privacy standards. Furthermore, we have open-sourced our dataset and model to facilitate further research in this area and to promote transparency and reproducibility.

We do not foresee any significant negative societal impacts arising from this work. However, we recognize that as with any security-related technology, there is potential for misuse, such as the development of adversarial techniques to bypass detection mechanisms. We encourage further research to strengthen anomaly detection models and ensure they are robust against such challenges.

Overall, our work is a step toward improving the security and trustworthiness of blockchain ecosystems, with broad potential societal benefits in safeguarding users and systems in decentralized finance.

# B Pseudo Algorithm of `BlockScan`

We present the pseudo algorithm of `BlockScan` in Algorithm 1 to help readers understand the workflow of `BlockScan`.

---

**Algorithm 1: Workflow of `BlockScan`**

---

**Require:** Benign transactions $\mathcal{D} = \{D_1, \ldots, D_N\}$, transactions to be predicted $\mathcal{P} = \{T_1, T_2, \ldots, T_M\}$, mask percentage $g$, detection mask percentage $g$, top-$s$ candidates, threshold $k$

**Ensure:** Top-$k$ transactions $\hat{\mathcal{P}}$ ranked by anomaly score

1: **Tokenization:**
2: Initialize tokenizer $\mathcal{T}$ with preserved address tokens and special tokens
3: Train subword tokenization on remaining data to generate the final token dictionary
4: Save tokenizer $\mathcal{T}$
5: **Training Phase:**
6: **for** each transaction $D_i \in \mathcal{D}$ **do**
7:     Tokenize $D_i$ using $\mathcal{T}$
8:     Randomly select $g$ tokens from $D_i$ and mask them: $D_i' = \text{Mask}(D_i, m)$
9:     Train the model $h$ to minimize the loss function:

$$\mathcal{L} = \sum_{i=1}^{N} \mathbb{E}_{D_i} \left[ \log P(D_i | D_i', h) \right]$$

10: **end for**
11: Save the trained model $h^*$
12: **Detection Phase:**
13: **for** each transaction $T_j \in \mathcal{P}$ **do**
14:     Tokenize $T_j$ using $\mathcal{T}$
15:     Randomly select $g\%$ of tokens and mask them: $T_j' = \text{Mask}(T_j, g)$
16:     Use the trained model $h^*$ to predict the top-$s$ tokens for each masked token position:

$$\hat{T}_j = \{\hat{t}_{i,1}, \hat{t}_{i,2}, \ldots, \hat{t}_{i,s} \text{ for } i = 1, \ldots, n\}$$

17:     Calculate the failed prediction ratio (abnormality score) for $T_j$:

$$\text{Score}(T_j) = \frac{1}{n} \sum_{i=1}^{n} \mathbb{I}(t_i \notin \{\hat{t}_{i,1}, \ldots, \hat{t}_{i,s}\})$$

18: **end for**

---

## C  Additional Details on Dataset

Here we provide additional details on the dataset used in our experiments.

**Dataset Statistics.**   Specifically, our Ethereum dataset consists of 3,383 benign transactions for training, 709 benign transactions for testing, and 10 malicious transactions. The data was collected from October 2020 to April 2023. For Solana, our training dataset comprises 35,115 transactions, while the testing dataset includes 1,500 benign transactions and 18 malicious transactions. The Solana data is sampled in September 2023 and December 2023, spanning a two-month period due to the availability of transaction data. To mitigate the risk of data leakage, we ensured that the malicious transactions occurred after the sampling periods of benign transactions. This approach guarantees that the model is trained solely on known benign transactions up to the cutoff dates, preventing any inadvertent exposure to future anomalous patterns during training.

**Address Frequency.**   To balance training efficiency and information retention, we rank all unique addresses in the dataset by frequency and retain only the top 7,000 addresses for training. For Ethereum, this covers the majority of unique addresses, as there are only 7,335 addresses in total, with the remaining addresses appearing just once in the training set. For Solana, the dataset contains 56,203 unique addresses, and retaining all of them would significantly increase the embedding size, making training computationally infeasible due to the high resource demands. Notably, the addresses excluded from the top 7,000 in Solana appear fewer than 10 times in the training set, contributing minimally to the overall information.

High-frequency addresses typically correspond to smart contracts, token addresses, or other entities that are frequently accessed and more significant for classification tasks. Conversely, low-frequency addresses, such as individual user wallets, often carry less relevance for anomaly detection. Including these low-frequency addresses would increase model complexity and training time without yielding significant performance gains. By focusing on the most frequent 7,000 addresses, we ensure a practical trade-off between training efficiency and the retention of critical information for effective anomaly detection.

**Potential Duplication in Transaction Data.**   Contract templates are wildly used in real-world smart contract development, leading to different smart contracts may offering similar or even identical APIs to the users. This could cause potential duplication in the transaction data. To assess the extent of this issue, we conduct a 5-gram BLEU similarity analysis on our dataset, choosing 5-gram to avoid false positives caused by indicator tokens such as "[START]" and "[CALL]." Our analysis reveal that only 0.05% of transaction pairs in the Ethereum dataset exhibit a BLEU similarity exceeding 0.7, with 0.023% surpassing 0.8. These highly similar transactions may indeed result from the use of contract templates. Given the low similarity ratio in our data, we do not consider potential duplication a significant issue.

## D  Detailed Experimental Results

### D.1  Implementation Details

**Our method**   We detail the hyper-parameters and training process of our customized language models, each trained from scratch for either the Solana or Ethereum tasks. Recall that for the Solana dataset, the model is based on a BERT-large architecture, with a hidden dimension of 1024, 24 hidden layers, and 16 attention heads. For the Ethereum dataset, the model uses a BERT-base architecture, with a hidden dimension of 768, 12 hidden layers, and 12 attention heads. The complete set of training hyper-parameters is detailed in Table 4 and Table 5. The Solana model was trained over two days using eight A100 GPUs, while the Ethereum model required around 2 hours of training on the same hardware.

**Baselines**   We employ four baseline methods: BlockGPT, Doc2Vec, GPT-4o, and Heuristic. For BlockGPT, as the source code was unavailable, we contact the author and implement BlockGPT based on their guidance. For the Doc2Vec approach, as described by [7], we first apply Doc2Vec [30] to extract features from the pre-processed and flattened traces of training transactions, as is shown in Figure 1. After obtaining the feature representations, we build a GMM to model the training

| config | value |
|---|---|
| optimizer | Adam [33] |
| base learning rate | 5e-5 |
| weight decay | 0.0 |
| gradient accumulation step | 10 |
| optimizer momentum | $\beta_1, \beta_2 = 0.9, 0.999$ |
| batch size | 3 |
| learning rate schedule | cosine decay |
| warmup epochs | 1 |
| total epochs | 10 |
| max sequence length | 8192 |

Table 4: **Configuration of training setup on Solana.**

| config | value |
|---|---|
| optimizer | Adam |
| base learning rate | 5e-5 |
| weight decay | 0.0 |
| gradient accumulation step | 10 |
| optimizer momentum | $\beta_1, \beta_2 = 0.9, 0.999$ |
| batch size | 20 |
| learning rate schedule | cosine decay |
| warmup epochs | 10 |
| total epochs | 100 |
| max sequence length | 1024 |

Table 5: **Configuration of training setup on Ethereum.**

transactions' distribution using the Sklearn library [34] with default hyper-parameters. During evaluation, for each transaction, we extract its feature using Doc2Vec and computed its anomaly score as the negative log-likelihood under the GMM. For the heuristic method, the anomalous score of a given transaction is determined by the sequence length of the corresponding flattened traces, with longer traces indicating a higher probability of anomaly behavior. For GPT-4o, we use the above prompts to instruct the LLM to give a score between 0 and 100. We use chain-of-thought (COT) [35] prompting to further improve the performance of GPT-4o. Additionally, we integrate human prior knowledge into the LLM by providing it with the list of known anomalous patterns to help it make more accurate predictions.

---

**Prompt for GPT-4o method**

You are a blockchain security expert tasked with determining whether a given blockchain transaction is anomalous. Please evaluate the transaction step by step and consider the following aspects:
1. Analyze the sender and recipient addresses to check if they have been involved in known anomalous activity.
2. Assess the transaction value and fee to identify any unusual patterns that might indicate suspicious behavior.
3. Examine the transaction's input data, including any smart contract interactions, to see if they match known attack vectors.
4. Consider the timing and frequency of the transaction relative to previous transactions from the same address.
Assign a score between 0 and 100, where 0 means completely benign and 100 means highly anomalous. Provide a clear explanation of the reasoning behind your score. Finally, return the result in the following JSON format:
#json
{ "reason": "Detailed explanation of why the transaction is considered anomalous or benign.", "score": "A number between 0 and 100 representing the likelihood of the transaction being anomalous."
}
Transaction details: [Insert transaction data here]

---

### D.2 Additional Experiments

**Post-Detection Methods.** As mentioned in §4.1, we also explore post-detection methods using a one-class contrastive learning approach. Specifically, we apply a SimSiam style [36] contrastive learning algorithm on top of the pre-trained BERT backbone to obtain more robust representation of the transactions. We then apply kernel density estimation (KDE) on the resulting representations to determine which transactions are anomalous. We provide a detailed explanation of this method below.

**Feature Extraction.** We first take each transaction $T_i$ (from the Ethereum dataset in this experiment) and tokenize it using the trained tokenizer $\mathcal{T}$. We then feed the token sequence into our pre-trained BERT model, denoted as $\mathcal{M}^*$, to obtain a feature vector $\mathbf{e}_i$. In practice, we consider two ways to

| Method | k=5 | | | k=10 | | | k=15 | | |
|---|---|---|---|---|---|---|---|---|---|
| | FPR | Recall | Precision | FPR | Recall | Precision | FPR | Recall | Precision |
| \<CLS\>-CL | 0.28% | 30% | 60% | 0.28% | 80% | 80% | **0.85%** | **90%** | **60%** |
| Average-CL | 0.14% | 40% | 80% | 0.28% | 80% | 80% | 0.97% | 80% | 53.33% |
| BlockScan | **0%** | **50%** | **100%** | **0.28%** | **80%** | **80%** | 0.97% | 80% | 53.33% |

Table 6: **Performance comparison of different post-detection methods for Ethereum.**

| Model | k=5 | | | k=10 | | | k=15 | | |
|---|---|---|---|---|---|---|---|---|---|
| | FPR | Recall | Precision | FPR | Recall | Precision | FPR | Recall | Precision |
| **GPT-4o** | 0.28% | 30% | 37.5% | 0.98% | 30% | 23% | 1.55% | 40% | 21% |
| **GPT-4o-FT** | 0.28% | 30% | 37.5% | 0.98% | 30% | 23% | 1.55% | 40% | 21% |

Table 7: **Performance comparison of fine-tuned GPT-4o and GPT-4 on Ethereum for various $k$ values.**

derive $\mathbf{e}_i$:

$$\mathbf{e}_i^{(\text{CLS})} = \text{BERT}_{\mathcal{M}^*}(T_i)\Big|_{\langle\text{CLS}\rangle\text{-token}}, \quad \mathbf{e}_i^{(\text{avg})} = \frac{1}{|T_i|} \sum_{t \in T_i} \text{BERT}_{\mathcal{M}^*}(t),$$

where $\mathbf{e}_i^{(\text{CLS})}$ is the embedding at the \<CLS\> token position, and $\mathbf{e}_i^{(\text{avg})}$ is the average of all token embeddings in $T_i$. We refer to these two modes as \<CLS\>-CL and Average-CL, respectively.

**SimSiam Contrastive Learning.** To further refine these representations, we apply one-class contrastive learning approach to $\mathbf{e}_i$. Specifically, we consider all samples from the training set as positive examples, as they represent benign transactions, and our goal is to learn a unified representation of these transactions. Consequently, since no negative samples are involved, we adopt the SimSiam contrastive learning framework. The overall SimSiam objective for a positive pair $(i, j)$ is then:

$$\mathcal{L}_{\text{Siam}} = \frac{1}{2}\Big[ D\big(h(f(\mathbf{e}_i)), f(\mathbf{e}_j)\big) + D\big(h(f(\mathbf{e}_j)), f(\mathbf{e}_i)\big) \Big].$$

Here, $f$ is a projection head (a one-layer MLP) and $h$ is a prediction head (another one-layer MLP). $D$ is the negative cosine similarity.

**KDE-Based Anomaly Scoring.** After the one-class contrastive learning, we obtain a refined feature vector $f(\mathbf{e}_i)$ for each transaction. We then fit a *Kernel Density Estimation* model with RBF kernel parameter $\gamma$ on the training set $\{f(\mathbf{e}_i)\}$ to estimate the density of a given transaction $T_j$. Formally, we have

$$\text{KDE}_\gamma(T_j) = \frac{1}{\gamma} \log \left[ \sum_{f(\mathbf{e}_i)} \exp\big(-\gamma \|f(\mathbf{e}_j) - f(\mathbf{e}_i)\|^2\big) \right]$$

a lower density score for a transaction indicates a higher probability of $T_j$ being anomalous.

**Comparison with Our Masked Prediction Approach.** We evaluate two variants of this alternative pipeline:

- $\langle\text{CLS}\rangle$-CL, which uses the $\langle\text{CLS}\rangle$-token embedding as $\mathbf{e}_i$,
- Average-CL, which uses the average of token embeddings for $\mathbf{e}_i$.

As shown in Table 6, neither of these one-class contrastive learning methods yields stronger detection performance than our main masked prediction-based approach, BlockScan. Given the extra complexity and computational overhead of contrastive training plus KDE, our simpler masked reconstruction strategy remains preferable in practical settings. Therefore, BlockScan defaults to the *masked prediction* post-detection procedure, which is straightforward, lightweight, and empirically more effective.

Additional details on hyper-parameter settings (e.g., MLP dimensions, RBF kernel parameter $\gamma$) are provided in our anonymous repository at `https://github.com/nuwuxian/tx_fm` for reproducibility.

| Model | k=5 | | | k=10 | | | k=15 | | |
|---|---|---|---|---|---|---|---|---|---|
| | FPR | Recall | Precision | FPR | Recall | Precision | FPR | Recall | Precision |
| **No Noise** | **0%** | **50%** | **100%** | **0.28%** | **80%** | **80%** | **0.97%** | **80%** | **53.33%** |
| **With Noise** | 0.14% | 40% | 80% | 0.56% | 60% | 60% | 1.26% | 60% | 40% |

Table 8: **Performance comparison of models with and without noise for Ethereum for various $k$ values.**

| Dataset | Training Time (s) | | GPU Memory Usage (GB) | |
|---|---|---|---|---|
| | With FlashAttention | Without FlashAttention | With FlashAttention | Without FlashAttention |
| **Ethereum** | 7,042 | 9,415 | 41.5 | 78.4 |
| **Solana** | 170,210 | - | 79.4 | - |

Table 9: **Impact of FlashAttention on training time and GPU memory usage for Ethereum and Solana datasets.**

| Model | k=5 | | | k=10 | | | k=15 | | |
|---|---|---|---|---|---|---|---|---|---|
| | FPR | Recall | Precision | FPR | Recall | Precision | FPR | Recall | Precision |
| **DeBERTa** | 0% | 50% | 100% | 0.28% | 80% | 80% | 0.97% | 80% | 53.33% |
| **RoBERTa** | 0% | 50% | 100% | 0.28% | 80% | 80% | 0.97% | 80% | 53.33% |

Table 10: **Performance comparison of different base models on Ethereum for various $k$ values.**

**Fine-Tuning GPT-4o.** We fine-tune GPT-4 (version 2024-08-06) using the Ethereum dataset to evaluate its performance on domain-specific tasks following **BlockGPT**'s approach. However, our method deviated from traditional token-by-token iteration approaches due to the limitations of OpenAI's fine-tuning API, which supports only instruction-response style fine-tuning. Instead, we divide each benign transaction into two halves: the first half served as the input, and the second half as the target response. This method aimed to enable the model to predict transaction details. The error between the predicted and actual transaction is used as the anomaly score. We summarize the results in Table 7.

The results indicate no significant improvement over the default GPT-4o model. Several factors may contribute to this outcome:

- **Training Budget Constraints**: Our fine-tuning costs are approximately $950, limiting the number of training iterations.
- **Coarse-Grained Approach**: The half-half prediction strategy may not have captured the intricate details of transaction patterns.
- **Tokenization Challenges**: GPT-4o's default tokenization struggles with specific data types, such as blockchain addresses and numerical patterns, reducing its ability to learn precise representations.

To overcome these limitations, future efforts could include:

- Developing more fine-grained fine-tuning strategies.
- Exploring additional tools to preprocess blockchain-specific inputs, such as addresses and numbers.
- Leveraging models with customizable tokenization and greater control over training objectives.

**Robustness to Noise.** We intentionally modified the training data to include 50% of the malicious transactions while keeping the rest of the data unchanged for the Ethereum dataset. As shown in Table 8, the detection performance of `BlockScan` is still relatively good, achieving a recall of 60% for a detection threshold $k = 10$. These results demonstrate that our approach maintains robustness to a reasonable level of noise and inaccurate information in the data.

### D.3 Ablation Study

**Impact of FlashAttention.** To evaluate the impact of FlashAttention on the training efficiency and resource utilization of `BlockScan`, we conduct an ablation study on Ethereum and Solana datasets. The results are shown in Table 9.

The integration of FlashAttention significantly improves training efficiency by optimizing attention computation. For Ethereum, FlashAttention reduces the running time from 9,415 seconds to approximately 7,000 seconds, as shown in the table. Additionally, it nearly halves the GPU memory usage, enabling more efficient use of hardware resources.

For Solana, the impact of FlashAttention is even more pronounced. Without FlashAttention, the model cannot handle even a batch size of 1 on an 80GB A100 GPU due to memory constraints. With FlashAttention, the training process becomes feasible, allowing a batch size of 2 while maintaining memory efficiency.

These results highlight the critical role of FlashAttention in handling long sequences and enabling scalable training for large datasets without sacrificing detection performance. Also, enabling FlashAttention has no noticeable impact on the accuracy of the model for Ethereum, as it achieves the same detection performance as the model without FlashAttention. This aligns with its design goal of optimizing computational efficiency rather than altering model representations or outputs. This study demonstrates that FlashAttention is essential for enabling efficient training on long sequences and large datasets while maintaining detection performance.

**Impact of Base Model.** To ensure that the choice of the base model does not significantly influence the performance of our framework, we conducted additional experiments with alternative state-of-the-art BERT-like models, such as DeBERTa [32], on the Ethereum dataset. These models are selected for their outstanding ability in NLP tasks. As shown in Table 10, the results achieved with DeBERTa are consistent with those of RoBERTa. This validation confirms that our framework is agnostic to the specific choice of base model, offering flexibility in adapting to other transformer-based architectures. Future work may explore additional models to further generalize the framework's applicability.

**Larger Dataset.** We conduct another experiment by only using manually verified malicious transactions without cross-checking on Ethereum dataset to conduct the experiment. The larger dataset is formed by 100 attack transactions belonging to 69 vulnerable contracts. We follow BlockGPT to categorize them based on the number of transactions interacting with the contract. We set the threshold (k) as $\leq 0.01, \leq 0.1, \leq 0.5, \leq 1, \leq 10$ of the total number of transactions interacting with each contract, and Top-1, Top-2, Top-3. The results are shown in Figure 2. From the results, we can see that BlockScan achieves consistent better performance than BlockGPT, which validates the effectiveness of BlockScan in detecting anomalous transactions within a larger dataset.

(a) BlockGPT Performance

| Dataset Size (Total Attacks interacting with vulnerable contracts) | Percentage Ranking Alarm Threshold (%) | | | | | Absolute Ranking Alarm Threshold | | |
|---|---|---|---|---|---|---|---|---|
| | ≤ 0.01 | ≤ 0.1 | ≤ 0.5 | ≤ 1 | ≤ 10 | Top-1 | Top-2 | Top-3 |
| **0 - 99 txs (14 attacks)** | | | | | | | | |
| Detected (Percentage) | — | — | — | — | 8 (57.1%) | 6 (42.8%) | 7 (50.0%) | 9 (64.2%) |
| Average false positive rate | — | — | — | — | 8.1% | 0.0% | 1.0% | 1.7% |
| Average num. of false positives | — | — | — | — | 4.32 | 0.6 | 1.5 | 2.3 |
| **100 - 999 txs (11 attacks)** | | | | | | | | |
| Detected (Percentage) | — | — | 4 (36.4%) | 4 (36.4%) | 7 (63.6%) | 3 (27.3%) | 3 (27.3%) | 4 (36.4%) |
| Average false positive rate | — | — | 0.38% | 1.00% | 9.00% | 0.0% | 0.0% | 0.0% |
| Average num. of false positives | — | — | 1.48 | 3.59 | 39.8 | 0.7 | 1.7 | 2.6 |
| **1000 - 9999 txs (20 attacks)** | | | | | | | | |
| Detected (Percentage) | — | 6 (30.0%) | 8 (40.0%) | 8 (40.0%) | 14 (70.0%) | 5 (25.0%) | 5 (25.0%) | 6 (30.0%) |
| Average false positive rate | — | 0.02% | 0.50% | 1.00% | 9.50% | 0.0% | 0.0% | 0.0% |
| Average num. of false positives | — | 3.80 | 18.60 | 37.8 | 380.4 | 0.8 | 1.75 | 2.7 |
| **10000+ txs (24 attacks)** | | | | | | | | |
| Detected (Percentage) | 2 (8.3%) | 4 (16.7%) | 8 (33.3%) | 10 (41.7%) | 14 (58.3%) | 0 (0.0%) | 1 (4.2%) | 3 (12.5%) |
| Average false positive rate | 0.01% | 0.12% | 0.52% | 0.98% | 9.08% | 0.0% | 0.0% | 0.0% |
| Average num. of false positives | 2.25 | 19.47 | 96.58 | 194.46 | 1942.79 | 1.0 | 2.0 | 3.0 |
| **Overall (69 attacks)** | | | | | | | | |
| Detected (Percentage) | 2 (8.3%) | 10 (14.5%) | 20 (29.0%) | 22 (31.9%) | 45 (65.2%) | 14 (20.2%) | 16 (23.2%) | 24 (34.8%) |
| Average false positive rate | 0.02% | 0.08% | 0.50% | 0.95% | 9.50% | 0.0% | 0.0% | 0.0% |
| Average num. of false positives | 2.25 | 19.49 | 96.58 | 193.45 | 1942.79 | 0.8 | 1.7 | 2.7 |

(b) `BlockScan` Performance

| Dataset Size (Total Attacks interacting with vulnerable contract) | Percentage Ranking Alarm Threshold (%) | | | | | Absolute Ranking Alarm Threshold | | |
|---|---|---|---|---|---|---|---|---|
| | ≤ 0.01 | ≤ 0.1 | ≤ 0.5 | ≤ 1 | ≤ 10 | Top-1 | Top-2 | Top-3 |
| **0 - 99 txs (14 attacks)** | | | | | | | | |
| Detected (Percentage) | — | — | — | — | 12 (85.71%) | 9 (64.2%) | 12 (85.71%) | 12 (85.71%) |
| Average false positive rate | — | — | — | — | 7.42% | 0.0% | 0.8% | 1.4% |
| Average num. of false positives | — | — | — | — | 3.35 | 0.3 | 1.1 | 2.1 |
| **100 - 999 txs (11 attacks)** | | | | | | | | |
| Detected (Percentage) | — | — | 5 (45.5%) | 5 (45.5%) | 8 (72.73%) | 4 (36.4%) | 4 (36.4%) | 5 (45.5%) |
| Average false positive rate | — | — | 0.35% | 1.00% | 8.87% | 0.0% | 0.0% | 0.0% |
| Average num. of false positives | — | — | 1.45 | 3.54 | 39.0 | 0.6 | 1.6 | 2.5 |
| **1000 - 9999 txs (20 attacks)** | | | | | | | | |
| Detected (Percentage) | — | 8 (40.0%) | 10 (50.0%) | 10 (50.0%) | 16 (80.0%) | 7 (35.0%) | 7 (35.0%) | 7 (35.0%) |
| Average false positive rate | — | 0.01% | 0.48% | 0.96% | 9.05% | 0.0% | 0.0% | 0.0% |
| Average num. of false positives | — | 3.75 | 18.5 | 37.3 | 380.2 | 0.7 | 1.7 | 2.7 |
| **10000+ txs (24 attacks)** | | | | | | | | |
| Detected (Percentage) | 3 (12.5%) | 5 (20.9%) | 10 (41.67%) | 12 (50.0%) | 16 (66.7%) | 1 (4.16%) | 2 (8.32%) | 5 (20.83%) |
| Average false positive rate | 0.01% | 0.10% | 0.49% | 0.98% | 9.08% | 0.0% | 0.0% | 0.0% |
| Average num. of false positives | 2.20 | 19.41 | 96.54 | 193.45 | 1942.79 | 1.0 | 1.9 | 2.8 |
| **Overall (69 attacks)** | | | | | | | | |
| Detected (Percentage) | 3 (4.34%) | 13 (18.84%) | 25 (36.23%) | 27 (39.13%) | 52 (75.36%) | 21 (30.43%) | 23 (33.33%) | 29 (42.02%) |
| Average false positive rate | 0.01% | 0.08% | 0.42% | 0.97% | 9.07% | 0.0% | 0.0% | 0.0% |
| Average num. of false positives | 2.20 | 13.54 | 64.53 | 102.64 | 1063.36 | 0.7 | 1.7 | 2.7 |

Figure 2: **Performance comparison of BlockGPT and `BlockScan` on the larger dataset.**

