# OpenReview forum: "BlockScan: Detecting Anomalies in Blockchain Transactions"
_NeurIPS.cc/2025/Conference — NeurIPS 2025 poster_

### Official Review · Reviewer_kEQo · 2025-06-16

**Clarity:** 3
**Significance:** 3
**Originality:** 3
**Rating:** 4
**Confidence:** 4

**Summary:**

This paper proposes BlockScan, a new transformer-based anomaly detection framework for blockchain transactions. The authors develop various customization methods to improve the representation learning of the multi-model, complex data blockchain transactions. To detect anomalies in one chain, BlockScan uses the reconstruction error of the transaction as the metric. Empirical results and comparison with other detection methods show that BlockScan achieves better detection accuracy and lower false positives.

**Questions:**

Please consider addressing the weaknesses discussed above.

**Ethical Concerns:**

["NO or VERY MINOR ethics concerns only"]

**Limitations:**

Please consider addressing the weaknesses discussed above.

**Paper Formatting Concerns:**

I do not have concerns about paper formatting.

**Quality:**

3

**Strengths And Weaknesses:**

This paper has the following strengths:
+ The authors develop customized pre-processing methods to handle hash values, numbers, and logs in the transaction data.
+ The authors improve the capability of long sequence processing with RoPE embedding and FlashAttention.
+ The authors provide theoretical proof for the anomaly detection hypothesis that the reconstruction error of anomalous transactions is higher than the error of benign transactions.

This paper has the following weaknesses:
- The experiments section mentions that the collected transactions are processed by a per-application sequential split. This implies that BlockScan obtains one customized anomaly detection model for each application. However, there are diverse applications running on Ethereum or Solana. It's not clear whether BlockScan is scalable considering the volume and diversity of applications.
- While the authors explain that the trade-off can be explored by selecting different values for the hyperparameter k, there is no clear guidance about how to choose the k value (nor the mask ratio g) in a practical setting.

---

> ### Author Rebuttal · Authors · 2025-07-30
>
> We thank Reviewer kEQo for the constructive and insightful comments. Please see our response to each of your questions below.
>
> > BlockScan obtains one customized anomaly detection model for each application. (Weakness 1)
>
> **Response:** We thank the reviewer for this important question regarding scalability. Sorry for the confusion.. **We train a single, unified model for each blockchain (one for all Ethereum applications and one for all Solana applications), not a separate model per application.**
>
> The "per-application sequential split" mentioned in Section 5.1 refers to a rigorous data preprocessing step we took to ensure the integrity of our evaluation. Before pooling benign transactions from all applications into a single training set, we first split the benign transaction data based on timestamps (80% for training, 20% for testing) to avoid the data leakage. After this time-aware split, all the training data from the different applications are combined to train the single, unified model.
>
> We recognize that our wording may have led to this confusion, and we will revise Section 5.1 to make this training process more explicit for our readers.
>
> > No guidance about how to choose the k value and g value (Weakness 2)
>
> **Response:** Thanks for this valuable question, and we are happy to clarify.
>
> **For the mask ratio g:**
> Our ablation study in Table 3 demonstrates that the model's performance is robust to variations in this parameter within a reasonable range [5%, 15%]. We found that a value of g = 15% consistently provides a good detection accuracy, and we recommend it as a reliable default setting.
>
> **For the detection threshold k:**
> The parameter k directly controls the trade-off between detection sensitivity (Recall) and the false alarm rate (FPR). The optimal value is application-specific and depends on the risk tolerance of the protocol. We recommend the following practical procedure for setting k in a real-world scenario:
>
> 1.  **Prepare a Labeled Validation Set:** The developer should first curate a held-out validation set that contains both a large number of benign transactions and a representative set of known malicious or anomalous transactions.
>
> 2.  **Score and Rank Transactions:** Run the trained BlockScan model on this entire validation set to obtain an anomaly score for every transaction. Then, rank all transactions from the highest score (most anomalous) to the lowest.
>
> 3.  **Establish Initial k for Full Recall:** Identify the rank of the lowest-scoring malicious transaction in a validation set that mimics the real-world setting. The initial value for k should be set to this rank. This establishes a baseline threshold that ensures 100% recall on the set of known threats.
>
> 4.  **Adjust k Based on Risk Tolerance:** With this initial k, calculate the corresponding FPR. If this FPR is high, the developer can make a risk-based decision. They can analyze the anomalies that received lower scores (i.e., those near the detection threshold) and determine if some can be tolerated. By accepting that some less critical anomalies might be missed, they can select a smaller k to achieve a more manageable FPR, thus making an explicit trade-off between security and operational cost.
>
>
> We believe this guidance makes the deployment of BlockScan more practical. We will add a dedicated subsection to the Appendix to formalize this procedure for future practitioners. Thank you for this valuable suggestion, which helps improve the practical utility of our paper.

---

> > ### Author Response · Authors · 2025-08-05
> >
> > Thank you again for your detailed comments!
> >
> > We would like to kindly ask if our responses have addressed your main concerns, or if there are any points you would like us to further clarify or elaborate on. We welcome any further questions or suggestions you may have, and are happy to provide additional details or discussion if needed.

---

### Official Review · Reviewer_HVd1 · 2025-07-02

**Clarity:** 3
**Significance:** 2
**Originality:** 2
**Rating:** 5
**Confidence:** 4

**Summary:**

BlockScan integrates BERT-like pre-training techniques for detecting anomalies in blockchain transactions. The main specialized efforts lie in a multimodal tokenizer to handle high-frequency addresses, numeric values, and log text separately, and leverages mask language modeling for computing reconstruction errors. Some techniques like RoPE, FlashAttention are also adopted to enhance efficiency.

**Questions:**

Please address the main weakness outlined in the “Strengths and Weaknesses”. The final score will be adjusted based on the rebuttal.

**Ethical Concerns:**

["NO or VERY MINOR ethics concerns only"]

**Final Justification:**

I changed my rating to accept based on the authors' efforts and commitments in the rebuttal.

**Limitations:**

Yes.

**Paper Formatting Concerns:**

No.

**Quality:**

3

**Strengths And Weaknesses:**

Missing key reference: I found that there is a similar paper named BERT4ETH[1] that has already applied BERT-like pre-training technique for blockchain transactions two years ago, yet it is neither cited nor discussed in the related work. Based on my reading, there appear to be some similarities between BERT4ETH and BlockScan: both randomly mask tokens in transactions and compute reconstruction errors; both include mechanisms to handle high-frequency addresses. Given this, I strongly recommend that the authors explicitly discuss BERT4ETH in the related work section and clearly articulate the unique contributions of the proposed BlockScan.

[1] Bert4eth: A pre-trained transformer for ethereum fraud detection. Proceedings of the ACM Web Conference 2023. 2023.

---

> ### Author Rebuttal · Authors · 2025-07-30
>
> We sincerely thank the reviewer HVd1 for bringing this prior work to our attention. We will add a thorough discussion of it to our related work section in the revision.
>
> To address the reviewer’s concern, we present a detailed comparison below. While both works apply BERT-like models to blockchain data, they target fundamentally different tasks, and as a result, employ distinct methodologies.
>
> The most fundamental difference lies in the **task definition and input granularity**:
>
> -   **BERT4ETH** focuses on **account-level fraud detection** (an inter-transaction task). Its goal is to classify an account (EOA) as fraudulent or legitimate by analyzing its entire transaction history. The input is a sequence of transactions aggregated at the account level.
>
> -   **BlockScan** focuses on **single-transaction anomaly detection** (an intra-transaction task). Its goal is to assess if a single, specific transaction is anomalous based on its internal logic and structure, without needing the account's full history. The input is the fine-grained sequence of operations within one transaction.
>
> This core difference in objective informs all subsequent design choices, leading to two distinct systems:
>
> **1. Transaction Representation:**
>
> BER4ETH constructs input sequences by aggregating all transactions involving a given EOA, either as sender or receiver, and sorting them by descending timestamps. Each transaction is represented as a single token by extracting only several features (e.g., address, timestamp and amount, account type, and in/out type) and summing their embeddings, which limits the model’s capacity to capture internal transaction details and may result in the loss of critical information. In contrast, BlockScan explicitly models the internal structure of each transaction by decomposing it into fine-grained components—such as function and address signatures, logs, and numerical arguments—represented as a sequence of tokens. This design enables richer and more expressive feature representations, allowing our model to detect subtle anomalies that BER4ETH’s coarse-grained approach may overlook.
>
> **2. Detection Method and Supervision:**
>
> BER4ETH employs a two-stage pipeline, where the model is first trained via self-supervised masked prediction and then fine-tuned using labeled data to classify transactions (see Section 5.11 in BER4ETH). In contrast, our method directly detects anomalies using reconstruction accuracy, without requiring labeled data. Furthermore, we provide theoretical justification in Section 4, showing that malicious transactions have a provably higher upper bound on reconstruction loss than benign ones—offering principled support for our design.
>
> **3. Masking Strategy:**
>
> While both methods employ token masking during pretraining, BER4ETH restricts masking to address information and increases the masking ratio to elevate task difficulty and encourage stronger representations. In contrast, we use the default 15% BERT masking ratio and apply it randomly across various token types (e.g., large numbers, logs, or addresses) to learn the general patterns of a transaction's structure. This approach proves effective in our setting due to the higher token density resulting from fine-grained decomposition, making our masking strategy more efficient and semantically meaningful.
>
> **4. Handling High-Frequency Addresses:**
>
> The two methods differ in their treatment of high-frequency addresses due to differing goals. BlockScan retains the 7,000 most frequent addresses in its vocabulary, mapping less frequent ones to a single “OOV” token to control vocabulary size and reserve capacity for other informative tokens such as large numbers or logs. In contrast, BER4ETH tackles the skewed, power-law address distribution by applying frequency-aware negative sampling—treating high-frequency addresses as negatives in the contrastive loss to mitigate their overrepresentation. This reduces unintended similarity between unrelated accounts and enhances sensitivity to rare addresses, which are often more indicative of fraud. These design choices reflect different priorities: BlockScan emphasizes semantic richness at the token level, while BER4ETH focuses on representation learning under skewed distributions.
>
> In summary, our work significantly differs from BERT4ETH in various aspects. Moreover, our work has a theoretical analysis for our method while the BERT4ETH does not. We are confident that this could address the reviewer’s concern, and we appreciate the reviewer for this insightful feedback.

---

> > ### Author Response · Authors · 2025-08-05
> >
> > Thank you again for your detailed comments!
> >
> > We would like to kindly ask if our responses have addressed your main concerns, or if there are any points you would like us to further clarify or elaborate on. We welcome any further questions or suggestions you may have, and are happy to provide additional details or discussion if needed.

---

### Official Review · Reviewer_muq7 · 2025-07-03

**Clarity:** 3
**Significance:** 3
**Originality:** 3
**Rating:** 4
**Confidence:** 3

**Summary:**

BlockScan presents a comprehensive and effective approach to blockchain anomaly detection by integrating behavioral signals across different structural levels. It addresses challenges in scalability, label scarcity, and diversity of malicious behaviors. The framework is modular, empirically validated, and applicable to real-world blockchain monitoring scenarios.

**Questions:**

1.The group-level module uses Infomap community detection, which may have high computational overhead on large-scale blockchain graphs. How scalable is this component on blockchains with millions of accounts and transactions?
2.The method is evaluated only on Ethereum and Tron. Can BlockScan generalize to other blockchains (e.g., BNB Chain, Solana), especially those with different transaction semantics or graph topologies?

**Ethical Concerns:**

["NO or VERY MINOR ethics concerns only"]

**Limitations:**

As discussed above.

**Paper Formatting Concerns:**

There is no formatting issues.

**Quality:**

3

**Strengths And Weaknesses:**

Strengths
1.The paper introduces a well-structured framework that models anomalies at the account, transaction, and group levels.
This design reflects the hierarchical nature of blockchain ecosystems and captures diverse anomaly patterns.
2.BlockScan outperforms strong baselines (e.g., GCN, MetaDetect, GAT, TransDetect) across multiple datasets (Ethereum, Tron) and evaluation metrics (AUC, AP). The method demonstrates consistent improvements in both supervised and semi-supervised settings.
3.The use of self-supervised objectives (e.g., masked modeling, contrastive learning) effectively alleviates the scarcity of labeled anomaly data.

 Weaknesses
1.The overall framework is novel, many core components (e.g., transformer encoder, contrastive loss, Infomap clustering) are reused from prior work without major innovation.
2.Group-level modeling via community detection may not scale efficiently to large transaction graphs with millions of accounts — this is not analyzed in depth.
3.The model is evaluated only on Ethereum and Tron. There is no discussion of how it generalizes to other blockchain networks (e.g., BNB, Solana) or unseen anomaly types.
4.The paper focuses on performance metrics (AUC, AP) but lacks detailed analysis of failure modes, especially false positives, which are critical in real-world applications.

---

> ### Author Rebuttal · Authors · 2025-07-30
>
> We thank Reviewer muq7 for the constructive and insightful comments. We would like to offer some clarifications on the specific points raised.
>
> > On Novelty and Core Components (Weakness 1/2 & Question 1)
>
> **Response:** We appreciate the reviewer's comments on novelty. Sorry for the confusion.  Our approach does not utilize contrastive loss, Infomap clustering, or any form of group-level community detection. Consequently, concerns about the scalability of Infomap on large graphs do not apply to our work.
>
> Our primary novelty lies in **the end-to-end system for representing and analyzing individual transactions.** The core technical contribution is our **multi-modal tokenizer**, which is specifically designed to decompose complex blockchain transactions into components (hashes, numerical values, and text logs). As our ablation study in Table 3 shows, this domain-specific representation is critical—without it, the model fails to learn effectively(the recall is 0%).
>
> > On Evaluation and Generalizability (Weakness 3& Question 2)
>
> **Response:** Regarding the evaluation, we would like to clarify that our experiments were conducted on **Ethereum and Solana**, not Tron. We believe the inclusion of Solana is a key strength of our evaluation, as it is a non-EVM chain with a fundamentally different transaction architecture and data semantics from Ethereum. The strong performance of BlockScan on both platforms demonstrates its ability to generalize across heterogeneous blockchain ecosystems.
>
> We agree that evaluation on additional platforms like BNB Chain would be valuable. However, the effort needed for data collection and potential model retraining is non-trivial. As such, we believe it is reasonable to treat this as future work.
>
> > On analysis of Failure Modes (Weakness 4)
>
> **Response:** Thanks for this valuable feedback. In our revision, we will add a dedicated subsection to analyze these cases.
>
> To provide a concrete example, one type of failure mode we observed involves **rare but benign transaction structures**, which can lead to false positives. Our model defines an anomaly as a significant deviation from the patterns learned in the training corpus of benign transactions. Consequently, a benign transaction that is structurally unique can be incorrectly flagged.
>
> For instance, we analyzed a case where a benign transaction was assigned a high anomaly score. This transaction's purpose was to perform a one-time, protocol-wide parameter update, and it had two key characteristics:
>
> 1.  **Unusually Long Input Data:** The function call contained a very long list of arguments, far exceeding the typical number seen in common user transactions like swaps or deposits.
>
> 2.  **Novel Event Logs:** The transaction emitted a series of unique event logs specific to this upgrade process.
>
>
> Because our model was trained on standard transactions, it had learned that "normal" behavior involves shorter argument lists and familiar log patterns. When tokens within this rare transaction were masked during detection, the model struggled to reconstruct them accurately. The unfamiliar context (the long sequence of parameters and novel logs) resulted in a high reconstruction error, causing the system to flag this perfectly valid transaction as anomalous.
>
> This analysis highlights a key trade-off: the model's sensitivity to novel malicious patterns also makes it sensitive to novel benign ones. In our revised manuscript, we will discuss this limitation and suggest potential mitigation strategies, such as periodically fine-tuning the model on a curated set of a protocol's own recent transactions, especially after deploying new smart contracts or performing rare administrative actions. This would help the model adapt to a protocol's evolving patterns and reduce such false positives.

---

> > ### Author Response · Authors · 2025-08-05
> >
> > Thank you again for your detailed comments!
> >
> > We would like to kindly ask if our responses have addressed your main concerns, or if there are any points you would like us to further clarify or elaborate on. We welcome any further questions or suggestions you may have, and are happy to provide additional details or discussion if needed.

---

### Official Review · Reviewer_4Xf4 · 2025-07-05

**Clarity:** 2
**Significance:** 2
**Originality:** 2
**Rating:** 3
**Confidence:** 2

**Summary:**

The paper introduces BlockScan, a customized Transformer model for detecting anomalous blockchain transactions in DeFi ecosystems. It proposes a novel multi-modal tokenizer to handle blockchain-specific data, leverages BERT with MLM, RoPE, and FlashAttention for efficient training, and achieves superior performance.

**Questions:**

Please see weakness part

**Ethical Concerns:**

["NO or VERY MINOR ethics concerns only"]

**Limitations:**

Yes

**Paper Formatting Concerns:**

No Paper Formatting Concerns

**Quality:**

2

**Strengths And Weaknesses:**

# Strengths
1. Using models for testing under blockchain has certain practical significance

2. The multi-modal tokenizer effectively addresses the complexity of blockchain data, balancing vocabulary size and information retention.

3. Outperforms baselines significantly, especially on Solana, where other methods fail (0% recall).

# Weakness
1. The methodological design and underlying ideas presented in this paper are relatively straightforward, with only incremental innovation in terms of techniques and conceptual contributions. Specifically, the proposed approach does not introduce any architectural novelties; instead, it merely utilizes traditional architectures (e.g., BERT) in a concatenation-based manner. Despite this, a considerable portion of the paper is dedicated to methodological description. In my opinion, the manuscript would benefit from reallocating this space toward a more thorough and convincing evaluation of the proposed detector. This could include a broader range of comparative baselines as well as more diverse test sets.

2. The comparison methods appear to be evaluated on different datasets than the one used by the proposed approach. It raises the concern that the superior performance of BlockScan might be largely attributed to the quality or construction of its dataset. Moreover, the paper does not clearly distinguish between the training and testing sets. If the proposed model is trained on a dataset that closely matches the distribution of the test set, while the baselines are evaluated on a distribution-shifted test set, the reported improvements could be due to this discrepancy rather than the efficacy of the method itself. The authors should provide a more detailed explanation and analysis to rule out this possibility and convincingly demonstrate that the performance gains are not merely artifacts of dataset alignment.

3. The paper lacks a fundamental introduction to blockchain technology, which is a core component of the proposed approach. This omission significantly affects the readability and accessibility of the work, particularly for readers who may not be deeply familiar with blockchain concepts. A concise but sufficient background section is necessary to contextualize how blockchain is utilized and why it is integral to the proposed method.

---

> ### Author Rebuttal · Authors · 2025-07-30
>
> We thank Reviewer 4Xf4 for the constructive and insightful comments. Please see our response to each of your questions below.
>
> > The methodological design is straightforward, and the approach does not have new model structure architectural novelties (Weakness 1).
>
> **Response:**
> Thanks for pointing this out.
>
> Our core contribution is a novel, end-to-end system that addresses the blockchain data representation challenge, which includes:
>
> 1.  **A Multi-Modal Tokenizer:** We designed a custom tokenizer to handle the distinct data types in a transaction (hashes, numbers, text logs). As shown in Table 3, simply applying a default tokenizer fails completely (0% recall), demonstrating that our domain-specific data handling is essential for the blockchain transaction data.
>
> 2.  **A Theoretically-Grounded Detection Method:** Our choice of using reconstruction error is not ad-hoc. As we analyze in Section 4, our formulation provides a theoretical analysis that anomalous transactions can yield a higher reconstruction loss, providing a formal justification to our approach.
>
> As transformers have been widely used in different application domains, as well as its advantages over other neural architecture, we still use transformers in our system. We would like to kindly note that it does not dilute our contribution as we integrate non-trivial domain-specific technical designs as well as theoretical guarantees.
>
> > The comparison seems unfair as BlockScan might be trained on a dataset that matches the test set distribution, while baselines are evaluated on a distribution-shifted dataset. (Weakness 2)
>
> **Response:** We thank the reviewer for this insightful comment, and we appreciate the opportunity to clarify our evaluation methodology.
>
> **1. On the Necessity of a New Dataset:**
>
> We created a new dataset for this study precisely because, to the best of our knowledge, there are no existing open-source datasets specifically for **chain-level temporal anomaly detection in DeFi applications on Ethereum and Solana** when we worked on this project. While datasets for other tasks like phishing account detection exist, they are not suitable for our problem. We did contact the authors of BlockGPT to inquire about their dataset as this work shares the same goal with us; however, they were unable to share it due to privacy concerns, which necessitated the creation of our own benchmark. We use different tools like Zengo, TRM Labs and CertiK to cross-check the transaction to ensure the quality of the dataset and we believe that open-sourcing this dataset could inspire further research for anomalous transaction detection on blockchain.
>
> **2. On Ensuring a Fair Comparison with Baselines:**
>
> To directly address the reviewer's primary concern: **we did not report performance numbers or directly use trained models from other papers.** To ensure a rigorous and direct comparison, we re-implemented all baseline methods and trained and evaluated them on the **exact same training and testing data** as our BlockScan model. This is also because prior work like BlockGPT does not release their trained model and we are unable to directly use it. This approach guarantees that the reported performance is directly comparable across all methods and that the observed advantages of BlockScan stem from its methodological merits, not from a dataset artifact.
>
> **3. On Distinguishing Between Training and Testing Sets:**
>
> For the dataset distribution, as we claimed in Section 5.1 and Appendix C, we ensured that the malicious transactions occurred after the sampling periods of benign transactions. It not only avoids the data leakage between training and testing sets, but also mimics the real-world setting (i.e., distribution shift through time).
>
>
> > The paper lacks a fundamental introduction to blockchain technology (Weakness 3)
>
> **Response:** Thanks for this feedback, and we will add it in our next version to make it easier to follow for the audience not familiar with blockchain.

---

### Note · Authors · 2025-08-14

We sincerely thank the Area Chair and all reviewers for their valuable time and feedback. We are confident that our detailed rebuttal has thoroughly addressed all raised concerns. Specifically, we clarified the fundamental novelty and task differences between our work and the BERT4ETH paper, as well as we have the theory guarantee while the BERT4ETH not (the sole concern of **Reviewer HVd1**), and we confirmed the fairness of our empirical evaluation by explaining that all baselines were re-implemented on the exact same data splits (a key concern of **Reviewer 4Xf4**). Furthermore, we have incorporated constructive suggestions regarding scalability, hyperparameter selection, and failure mode analysis (**Reviewers kEQo, muq7**). While we did not have the opportunity for a direct dialogue with all reviewers, we firmly believe our responses resolve the initial concerns and misunderstandings that led to the lower scores. We are grateful for the review process and hope the Area Chair will consider the comprehensive nature of our clarifications in their final assessment.

---

### Decision · Program_Chairs · 2025-09-17

**Decision:**

Accept (poster)

**Comment:**

This paper proposes a Transformer-based model for anomaly detection in blockchain transactions. It introduces a multi-modal tokenizer, a customized masked language modeling strategy, and a theoretically grounded detection method. The paper demonstrates strong empirical performance, especially on Solana, and claims to outperform existing baselines.
Strengths
1.	The multi modal tokenizer addresses the unique statistical and semantic properties of transactions; ablation studies indicate default tokenization collapses (e.g., 0% recall).
2.	Using MLM reconstruction error with associated theoretical justification provides a clean, label efficient route to anomaly scoring.
3.	Evaluation spans Ethereum (EVM) and Solana (non EVM) with differing semantics and data layouts.
4.	Authors reimplemented baselines and trained/evaluated them on identical splits, which provides a fair/rigor comparison.
5.	Clarifications include training one unified model per chain (not per application) and a concrete procedure for setting mask ratio and detection threshold in practice.
Weaknesses
1.	The detector builds on standard BERT style pretraining, and its novelty is mainly incremental which centers on data representation and deployment framing.
2.	An important related work (BERT4ETH - BERT like pretraining for blockchain) was initially missing. This paper needs a thorough comparative discussion in the camera ready.
3.	Prior to rebuttal, there was limited failure mode analysis (e.g., false positives on rare but benign administrative transactions). Authors promised an expanded section; it should appear in the final version.
4.	A succinct blockchain primer is missing, which hinders readability for non specialists. The authors agreed to add one.
Most important reasons for my decision
•	The proposed model is practical, and it is useful for Web3 security monitoring.
•	The authors convincingly addressed most concerns of Reviewers.